**Data Availability Statement:** All newly generated DNA sequences for this study were submitted to GenBank under accession numbers MT179848 – MT179854 for 18S sequences; MT472183 –

# Phylogenetic evidence for the invasion of a commercialized European *Phasmarhabditis hermaphrodita* lineage into North America and New Zealand

**Dana K. Howe**[1]*, **Anh D. Ha**[1], **Andrew Colton**[2], **Irma Tandingan De Ley**[3], **Robbie G. Rae**[4], **Jenna Ross**[5,6,7], **Michael Wilson**[8], **Jiří Nermut**[9], **Zhongying Zhao**[10], **Rory J. Mc Donnell**[2], **Dee R. Denver**[1]

**1** Department of Integrative Biology, Oregon State University, Corvallis, Oregon, United States of America, **2** Department of Crop and Soil Science, Oregon State University, Corvallis, Oregon, United States of America, **3** Department of Nematology, University of California-Riverside, Riverside, California, United States of America, **4** School of Biological and Environmental Sciences, Liverpool John Moores University, Liverpool, United Kingdom, **5** Department of Conservation Ecology and Entomology, Faculty of AgriSciences, Stellenbosch University, Matieland, South Africa, **6** Institute of Biological and Environmental Sciences, University of Aberdeen, Aberdeen, United Kingdom, **7** Crop Health and Protection (CHAP), National Agri-Food Innovation Campus, Sand Hutton, York, United Kingdom, **8** Independent Researcher/Consultant, Hamilton, New Zealand, **9** Biology Centre CAS, Institute of Entomology, Branišovská, České Budějovice, Czech Republic, **10** Department of Biology, Hong Kong Baptist University, Hong Kong SAR, China

* howeda@oregonstate.edu

## Abstract

Biological control (biocontrol) as a component of pest management strategies reduces reliance on synthetic chemicals, and seemingly offers a natural approach that minimizes environmental impact. However, introducing a new organism to new environments as a classical biocontrol agent can have broad and unanticipated biodiversity effects and conservation consequences. Nematodes are currently used in a variety of commercial biocontrol applications, including the use of *Phasmarhabditis hermaphrodita* as an agent targeting pest slug and snail species. This species was originally discovered in Germany, and is generally thought to have European origins. *P. hermaphrodita* is sold under the trade name Nemaslug®, and is available only in European markets. However, this nematode species was discovered in New Zealand and the western United States, though its specific origins remained unclear. In this study, we analyzed 45 nematode strains representing eight different *Phasmarhabditis* species, collected from nine countries around the world. A segment of nematode mitochondrial DNA (mtDNA) was sequenced and subjected to phylogenetic analyses. Our mtDNA phylogenies were overall consistent with previous analyses based on nuclear ribosomal RNA (rRNA) loci. The recently discovered *P. hermaphrodita* strains in New Zealand and the United States had mtDNA haplotypes nearly identical to that of Nemaslug®, and these were placed together in an intraspecific monophyletic clade with high support in maximum likelihood and Bayesian analyses. We also examined bacteria that co-cultured with the nematode strains isolated in Oregon, USA, by analyzing 16S rRNA sequences. Eight different bacterial genera were found to associate with these nematodes, though *Moraxella osloensis*, the bacteria species used in the Nemaslug® formulation, was not detected.

MT472270 for mtDNA sequences (https://www.ncbi.nlm.nih.gov/nucleotide/).

**Funding:** This work was supported by an Oregon Department of Agriculture Nursery Research grant ODA 18-06 and an Agricultural Research Foundation grant awarded to RJM and DRD. https://www.oregon.gov/ODA/programs/NurseryChristmasTree/Pages/Grants.aspx https://agresearchfoundation.oregonstate.edu/grant-program Part of this research was also funded by the California Department of Food and Agriculture Specialty Crop Program (CDFA) grants SCB12059 (2012) and 29512 (2015) awarded to ITDL, and the Protein Research Foundation (Trust 5802/94 – Project P07/20/179/14) and by the National Research Foundation of South Africa (NRF-THRIP TP14062571871) awarded to JR. https://www.cdfa.ca.gov/specialty_crop_competitiveness_grants/ https://www.proteinresearch.net/ https://www.nrf.ac.za/ The funders had no role in study design, data collection and analysis, decision to publish, or preparation of the manuscript.

**Competing interests:** The authors have declared that no competing interests exist

This study provided evidence that nematodes deriving from the Nemaslug® biocontrol product have invaded countries where its use is prohibited by regulatory agencies and not commercially available.

## Introduction

Biological control, 'biocontrol', is a human-mediated strategy used to suppress populations of pests and other undesirable species that involves the active deployment or conservation of natural enemies of the target species. Biocontrol strategies are common components of integrated pest management (IPM) programs, and are commonly deemed more favorable than chemical-based and other pest control strategies where environmental damage and potential human health impacts are matters of concern [1–3]. The commercialization of biocontrol organisms started in the 1920s with development of *Encarsia formosa*, a parasitoid wasp species, as an agent used to combat greenhouse whiteflies [4]. Biocontrol products have diversified and increased over recent decades, expanding the overall abundance of these organisms and the geographic range across which they occur.

Although biocontrol strategies are often viewed favorably because they offer more natural approaches as compared to those involving chemicals, biodiversity and conservation concerns are also prominent components of the biocontrol dialogue. In particular, biocontrol approaches that involve the importation of natural enemies to locales where they did not previously exist can have profound unintended ecosystem effects. Introduced species intended for biocontrol applications can negatively impact native, non-target species; most of the well-known cases involve vertebrates. One classic example involved the misguided introduction of the small Indian mongoose, *Herpestes javanicus*, to Hawaii for the purpose of controlling rats. The mongoose, however, were discovered to be opportunistic feeders that preyed on the eggs and hatchlings of native ground nesting birds, and endangered sea turtles of Hawaii [5,6]. Impacts of introduced species can also be less direct, though still devastating to affected ecosystems. The Hawaiian cane toad, *Rhinella marina*, was intentionally introduced to Australia in 1935 to control greyback cane beetles, *Dermolipida albohirtum*, that were a pest in sugarcane production systems. The introduced toads were unable to reach the cane beetles, which resided high on cane stalks beyond the reach of the toads, so instead fed on other insects and rapidly spread across much of northeastern Australia during the subsequent decades. The cane toads took over habitats used by native amphibian species and brought new diseases to the native toads and frogs of Australia that dramatically reduced their populations [7]. Further, the poison produced by the cane toads negatively impacted populations of native predators such as goannas and tiger snakes. The predatory snail *Euglandina rosea* has been introduced to Pacific and Indian Ocean islands for the purpose of controling giant African land snails (*Lissachatina fulica*), but has turned out to be ineffective against the target invasive species and damaging to island-endemic gastropods instead [8]. These classic 'biocontrol gone awry' and similar stories have motivated the development of rigorous regulatory policies surrounding efforts to develop new biocontrol initiatives, requiring careful environmental impact analyses and other forms of scientific scrutiny prior to biocontrol agent deployment [9–12].

Invertebrates such as parasitic nematode species have also been deployed as biocontrol agents and components of IPM programs, usually targeting insect pests. For example, entomopathogenic nematodes in two genera, *Steinernema* and *Heterorhabditis*, parasitize broad ranges of insect hosts and are commonly used to target scarab beetles and weevils [13,14].

These nematodes rely on obligate symbioses with mutualistic bacteria (*Xenorhabdus* spp. bacteria associate with *Steinernema* spp. nematodes; *Photorhabdus* spp. bacteria associate with *Heterorhabditis* spp. nematodes) to feed on their insect hosts. *Deladenus siricidicola* offers a second example of a nematode species used in insect biocontrol; this nematode infects *Sirex noctilio*, a species of woodwasp which causes significant tunnelling damage to the trunks of pine trees [15,16]. *D. siricidicola* infection results in sterilized female wasps, which also vector the nematodes to nearby trees where they can infect new hosts.

*Phasmarhabditis hermaphrodita* is a nematode species commercialized for biocontrol applications in Europe [17–19]. Unlike the previously discussed biocontrol nematodes, *P. hermaphrodita* parasitizes a variety of pest slug and snail species; the most common biocontrol target is the grey field slug, *Deroceras reticulatum*. *P. hermaphrodita* is grown in mass production with a bacteria, *Moraxella osloensis* used as a food source, and sold under the tradenames Nemaslug® and SlugTech®. The relative roles of the nematode and bacteria in causing slug host mortality remain unclear, though the ability of *P. hermaphrodita* to grow and reproduce by feeding on many different types of bacteria in the lab and other recent findings [20–22] suggest that any putative bacterial symbioses in this system must be facultative in nature. Other species in this nematode genus, such as *Phasmarhabditis papillosa* and *Phasmarhabditis californica*, have been proposed as potential additional or alternative biocontrol agents [23,24], though *P. hermaphrodita* offers the only currently available nematode biocontrol product targeting pest slugs and snails.

*P. hermaphrodita* was originally discovered in Germany [25] and then later in France [26] and is often assumed to be of European origin. The *P. hermaphrodita* strain that became Nemaslug® was isolated in Bristol, England [18]. With increased interest in this potential biocontrol there has been increased surveying efforts; *P. hermaphrodita* has also been discovered in Chile [27], Iran [28], Egypt [29], and more recently in New Zealand [30,31] and the west coast of North America [32,33]. It remains unclear, however, whether these reported observations of *P. hermaphrodita* in non-European locations constitute previously unknown natural populations of the species, or instead resulted from the introduction of Nemaslug®. The latter possibility would pose a matter of strong conservation and regulatory concern. Nemaslug® is not licensed for sale and has never been sold in the United States or New Zealand, and the potential impacts of *P. hermaphrodita* on native gastropod species remains unknown.

In this study, we developed and applied a molecular genetic strategy to investigate the origins and diversity of *P. hermaphrodita* lineages obtained from different parts of the world. Our approach, based on rapidly evolving mitochondrial DNA (mtDNA) sequences, was further applied to additional *Phasmarhabditis* spp. to provide broader insights into evolutionary relationships in this group of nematodes, and guide tree rooting decisions for phylogenetic analyses focused on *P. hermaphrodita*. We also examined the bacterial associates of three *P. hermaphrodita* isolates found in Oregon to add knowledge on the potential relationships between this nematode species and putative bacterial symbionts. Our study demonstrated that mtDNA is a powerful tool for investigating within- and between-species evolutionary patterns in this nematode group, found variable bacteria associated with the three different *P. hermaphrodita* isolates from Oregon, and revealed strong evidence that recently discovered *P. hermaphrodita* isolates from California, Oregon, and New Zealand all likely derived from Nemaslug®.

## Materials and methods

### Nematode populations

Nematodes were isolated from slugs collected from field and nursery sites in Oregon. Briefly, collected slugs were maintained in a laboratory growth chamber at 16.5°C and 12:12 light in

16oz plastic containers (9cm diameter), corresponding to the site of origin. A maximum of 20 slugs were placed in each container to avoid stress or cross contamination. Each container was kept damp with deionized water-soaked towels, and the slugs were fed organic carrot. Dead slugs, and slugs showing signs of illness, were removed and placed in a petridish (5cm diameter) for further monitoring. Upon death, all slugs were visually inspected using a Leica microscope for the presence of nematodes. If detected, nematodes were washed off the slug carcass using M9 buffer and transferred to standard NGM plates for culturing. After 7–10 days, nematodes were collected from the plates for DNA extraction and sequencing. Additional *Phasmarhabditis* species were isolated from various locations around the globe (Table 1). These nematodes were preserved in DESS (a solution of dimethyl sulphoxide, disodium EDTA, and saturated NaCl) [34] and shipped to Oregon for DNA extraction and sequencing. Most locations did not require permits for slug collection, however, the slugs collected in California fell under California Department of Food and Agriculture (CDFA) permit 2942 (October 2012 to August 2014).

## PCR and DNA sequencing

Nematodes were collected in a proteinase-K based lysis buffer for DNA extraction as previously described [35]. New nematode isolates' species were identified using 18S ribosomal DNA amplification and direct sequencing, and collaborator nematode samples were verified using 18S rRNA amplification and direct sequencing [36]. All 18S rRNA sequences were compared to GenBank's nr database using blastn. When the BLAST match was a *Phasmarhabditis* species (percent identity of 99–100), the population was incorporated into this mtDNA study.

PCR amplification of ~5 kb of mtDNA was achieved using the Expand Long Range PCR kit (Roche) as previously described [37]. The mtDNA was targeted with ribosomal primers designed to match across a wide range of rhabditid nematodes. This segment of mtDNA included the genes: rrnL, nad3, nad5, tRNA-Ala, tRNA-Pro, tRNA-Val, nad6, nad4L, tRNA-Trp, tRNA-Glu, and rrnS (S1 Fig). Direct end sequencing of the large mtDNA products were generated from all *Phasmarhabditis* species. For *P. hermaphrodita* and *Phasmarhebditis neopapillosa* samples, subsequent sequencing of the large mtDNA products was performed using newly designed genus-specific primers (S1 Table). All newly generated DNA sequences for this study were submitted to GenBank under accession numbers MT179848 –MT179854 for 18S sequences and MT472183 –MT472270 for mtDNA sequences.

## Phylogenetic analysis

Two phylogenetic analyses were performed using two different mtDNA sequence data sets. The first analysis targeted the relationships within the broader genus *Phasmarhabditis* and included 45 nematode strains from eight species, along with three nematode outgroup strains: *Pristionchus pacificus* NC_015245 [38], *Oscheius chongmingensis* KP257594 and *Caenorhabditis elegans* NC_001328 [39]. The second analysis explored the relationship within *P. hermaphrodita* more closely using a longer region of mtDNA sequences from 24 nematode strains collected from four countries and four *P. neopapillosa* strains, the nearest outgroup. MEGA7 was used for sequence alignment and phylogenetic model testing of all data sets [40].

For the first analysis, the best fit model for the comparison of 1,121 bp of mtDNA from multiple *Phasmarhabditis* species was the Hasegawa-Kishino-Yano model, with Gamma distribution and invariable sites (HKY+G+I) [41]. This model was used to generate a maximum-likelihood molecular evolution phylogeny with 1,000 bootstrap replicates in MEGA7. A Bayesian inference phylogeny was also generated with model HKY85+G+I using the MrBayes [42] plugin in Geneious Prime 2019.1.3 (https://www.geneious.com).

**Table 1.** *Phasmarhabditis* spp. strains included in this study.

| Strain ID | Species | Contributor[a] | Collection Date | Collection Location | Host |
|---|---|---|---|---|---|
| BAR | *P. apuliae* | BC CAS | November 2012 | Bari, Italy | *Milax sowerbyi* |
| MGS | *P. bonaquaense* | BC CAS | May 2018 | Stakcin, Slovakia | *Limax cinereoniger, L. maximus* |
| DL305 | *P. californica* | OSU | April 2018 | Brookings, Oregon | *Deroceras reticulatum* |
| DL310 | *P. californica* | OSU | June 2018 | Eugene, Oregon | *Deroceras reticulatum* |
| ITD046 | *P. californica* | UCR | January 2013 | McKinleyville, California | *Deroceras reticulatum* |
| ITD725 | *P. californica* | UCR | August 2014 | Eureka, California | *Deroceras laeve* |
| ITD727 | *P. californica* | UCR | August 2014 | Eureka, California | *Deroceras laeve* |
| DMG018 | *P. californica* | LJMU | April 2014 | Pembrokeshire, Wales | *Oxychilus draparnaudi* |
| DMG019 | *P. californica* | LJMU | April 2014 | Pembrokeshire, Wales | *Oxychilus draparnaudi* |
| NZPc | *P. californica* | Wilson | June 2014 | Tokoroa, Waikato | *Deroceras reticulatum* |
| DL300 | *P. hermaphrodita* | OSU | March 2017 | Corvallis, Oregon | *Deroceras reticulatum* |
| DL307 | *P. hermaphrodita* | OSU | February 2019 | Salem, Oregon | *Deroceras reticulatum* |
| DL309 | *P. hermaphrodita* | OSU | February 2019 | Salem, Oregon | *Deroceras reticulatum* |
| NZPh | *P. hermaphrodita* | Wilson | June 2014 | TeMata Peak, Hawke's Bay | *Deroceras invadens* |
| ITD056 | *P. hermaphrodita* | UCR | February 2013 | Sonoma, California | *Deroceras laeve* |
| ITD207 | *P. hermaphrodita* | UCR | January 2013 | Eureka, California | *Deroceras reticulatum* |
| ITD272H2-1 | *P. hermaphrodita* | UCR | January 2013 | Eureka, California | *Deroceras reticulatum* |
| ITD290B113 | *P. hermaphrodita* | UCR | January 2013 | Eureka, California | *Lehmannia valentiana* |
| ITD803-2 | *P. hermaphrodita* | UCR | May 2016 | Oakland, California | *Deroceras reticulatum* |
| KNAH3A | *P. hermaphrodita* | UCR | November 2018 | Tehama, California | *Arion hortensis* agg. |
| Nemaslug | *P. hermaphrodita* | N/A | 1993 | Bristol, England | *Deroceras reticulatum* |
| DMG002 | *P. hermaphrodita* | LJMU | January 2014 | Liverpool, England | *Arion* spp. |
| DMG003 | *P. hermaphrodita* | LJMU | January 2014 | Liverpool, England | *Deroceras panormitanum* |
| DMG005 | *P. hermaphrodita* | LJMU | January 2014 | Liverpool, England | *Arion subfuscus* |
| DMG006 | *P. hermaphrodita* | LJMU | January 2014 | Liverpool, England | *Arion subfuscus* |
| DMG007 | *P. hermaphrodita* | LJMU | January 2014 | Liverpool, England | *Limax flavus* |
| DMG008 | *P. hermaphrodita* | LJMU | February 2014 | Liverpool, England | *Deroceras panormitanum* |
| DMG009 | *P. hermaphrodita* | LJMU | February 2014 | Liverpool, England | *Deroceras panormitanum* |
| DMG010 | *P. hermaphrodita* | LJMU | February 2014 | Liverpool, England | *Milax budapestensis* |
| DMG011 | *P. hermaphrodita* | LJMU | February 2014 | Liverpool, England | *Milax budapestensis* |
| B1 | *P. hermaphrodita* | CAS | June 2009 | Benesov u Prahy, Czechia | *Deroceras reticulatum* |
| CB1 | *P. hermaphrodita* | CAS | June 2009 | Ceske Budejovice, Czechia | *Deroceras reticulatum* |
| ROZ | *P. hermaphrodita* | CAS | June 2010 | Ceske Budejovice, Czechia | *Deroceras reticulatum* |
| DER | *P. hermaphrodita* | CAS | June 2011 | Ceske Budejovice, Czechia | *Deroceras reticulatum* |
| ZZY0412 | *P. huizhouensis* | HKBU | October 2013 | Huizhou, China | rotting leaves |
| DMG012 | *P. neopapillosa* | LJMU | January 2014 | Aberdeen, Scotland | *Deroceras reticulatum* |
| DMG014 | *P. neopapillosa* | LJMU | January 2014 | Liverpool, England | *Limax flavus* |
| DMG015 | *P. neopapillosa* | LJMU | January 2014 | Liverpool, England | *Limax flavus* |
| DMG016 | *P. neopapillosa* | LJMU | January 2014 | Liverpool, England | *Limax flavus* |
| SA4 | *P. papillosa* | Ross | October 2015 | George, South Africa | *Deroceras reticulatum* |
| ITD510 | *P. papillosa* | UCR | July 2013 | San Diego, California | *Deroceras reticulatum* |
| DL306 | *P. papillosa* | OSU | August 2018 | Portland, Oregon | *Deroceras reticulatum* |
| DL308 | *P. sp.* | OSU | February 2019 | Salem, Oregon | *Deroceras reticulatum* |
| DF5056 | *P. sp.* | UCR | May 1993 | Bronx, New York | earthworm |
| EM434 | *P. sp.* | UCR | April 1990 | Bronx, New York | earthworm |

[a] Contributor designations are BC CAS = Biology Centre CAS; HKBU = Hong Kong Baptist University; LJMU = Liverpool John Moores University; OSU = Oregon State University; Ross; UCR = University of California, Riverside; Wilson; N/A = not applicable.

For the second analysis, the phylogeny of 2,372 bp of *P. hermaphrodita* mtDNA, the best fit model was the Hasegawa-Kishino-Yano model, with Gamma distribution. A HKY+G maximum-likelihood evolution phylogeny with 1,000 bootstrap replicates was generated in MEGA7. A Bayesian inference phylogeny was also generated with model HKY85+G using the MrBayes plugin in Geneious Prime.

## Isolation and genetic characterization of bacteria

Nematodes of three Oregon *P. hermaphrodita* strains (DL300, DL307, and DL309) were cultured on NGM plates until a bacterial lawn established. Bacteria were then streaked onto a series of sterile LB nutrient agar plates to obtain single colonies. Twelve colonies from each nematode strain were collected in molecular biology-grade water (product number). DNA was isolated by incubating at 100°C in 10 minutes then immediately cooling on ice for 10 minutes.

Identification of bacteria was performed using 16S ribosomal DNA amplification and direct sequencing. Sequences of length ~1,460 bp in the 16S gene were amplified using the universal primers 27F and 1492R [43,44] (sequences: 27F (5′-AGAGTTTGATCCTGGCTCAG-3′) and 1492R (5′-GGTTACCTTGTTACGACTT-3′)). PCR products were then purified using ChargeSwitch PCR Cleanup Kit (Thermo Fisher Scientific). DNA sequences were determined by Sanger method at the Center for Genome Research and Biocomputing at Oregon State University. Sequences were compared to the NCBI nr database using BLASTN, and the first match with percent identity of $\geq$ 97% were reported as sequence ID.

## Results

### Phylogenetics of mtDNA in *Phasmarhabditis* spp

For the first analysis, a maximum likelihood with bootstrap analysis approach was used to phylogenetically analyze 1,121 bp of mtDNA from 48 nematode strains from eight different *Phasmarhabditis* species (Fig 1). The Bayesian inference analysis yielded a tree with an identical topology as maximum likelihood, therefore the results were combined. All *Phasmarhabditis* species formed a monophyletic group with strong support (99/1.00). The mtDNA sequences provide strong support, in both analysis methods, for the previously defined species. For six of the species, we analyzed multiple nematode mtDNA samples, and in every case these species were represented as unique monophyletic groups. The unnamed species, first discovered in earthworms and recently now found in slugs, is a strongly supported internal clade of *Phasmarhabditis*. Using mtDNA sequence, *P. hermaphrodita* is most closely related to *P. neopapillosa* and most distantly related to *Phasmarhabditis huizhouensis*.

### Phylogenetics of mtDNA within *P. hermaphrodita*

For the second analysis, additional mtDNA sequencing, for a total of 2,372 bp, was used in a maximum likelihood with bootstrap analysis approach to phylogenetically analyze 24 populations of *P. hermaphrodita*, with four populations of *P. neopapillosa* as an outgroup (Fig 2). A Bayesian analysis yielded an identical topology as that of maximum likelihood, therefore the results were combined. Within the *P. hermaphrodita* clade, there was strong support for two major subclades, defined here as Ph_mtI (95/0.84) and Ph_mtII (100/1.00). Subclade Ph_mtI contained the Nemaslug® strain, originally isolated in Europe, and all nematodes recently isolated from New Zealand and the United States of America; with one and nine strains, respectively. Ph_mtII contains only strains isolated from Europe, both England and the Czech Republic; nine and four strains respectively.

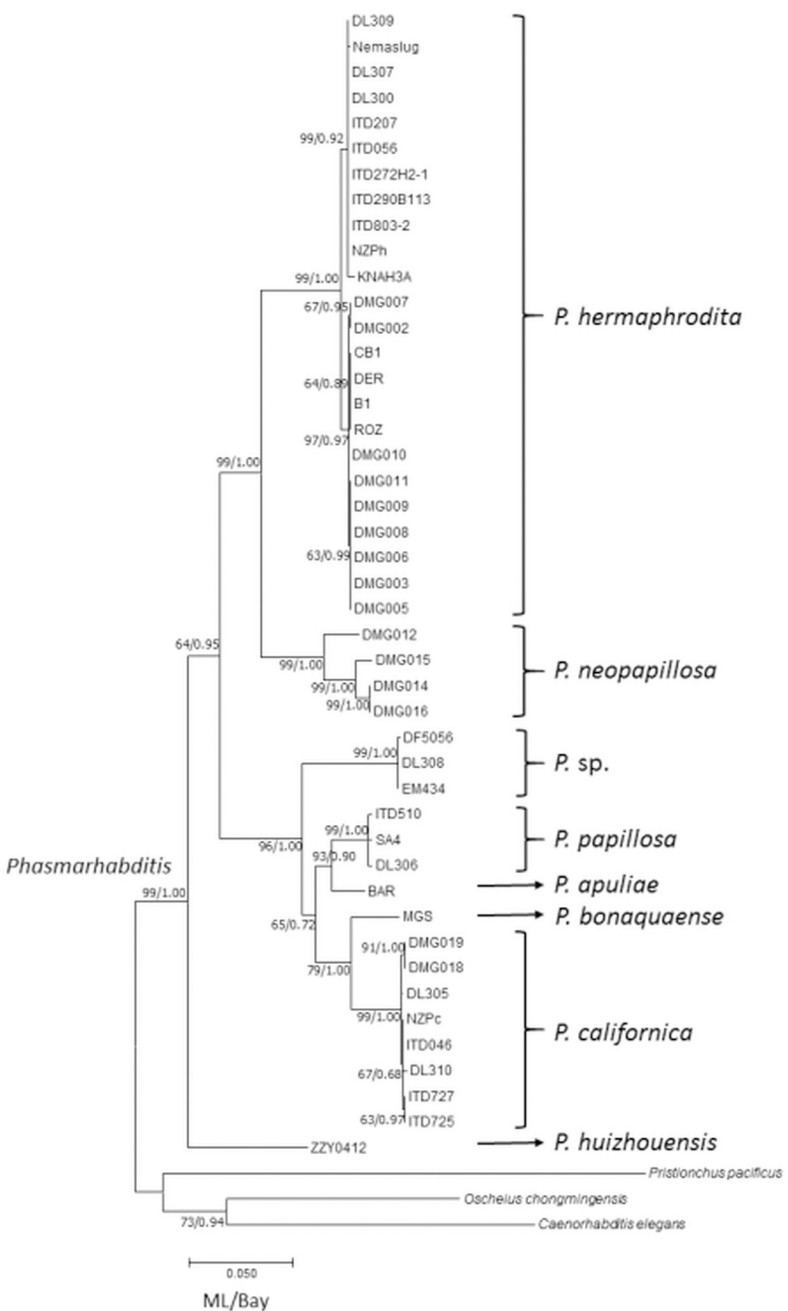

**Fig 1. Maximum likelihood phylogeny of ~1,120 bp of mtDNA from *Phasmarhabditis* species.** A total of 48 nematode mtDNA sequences, representing 8 *Phasmarhabditis* species and 3 outgroups, were analyzed. Additional information about the nematode populations is found in Table 1. Maximum likelihood and Bayesian phylogenies yielded identical topologies; node support lists maximum likelihood values on left (ML) and Bayesian posterior probabilities are on the right (Bay). The scale bar represents 0.050 substitutions per site.

## Bacterial associates of *P. hermaphrodita* from Oregon

Twelve bacterial colonies found in association with each of the three Oregon *P. hermaphrodita* were examined (Fig 3). Out of 12 colonies from the strain DL300, four were identified as bacteria of the genus *Sphingobacterium* (33%), and the other eight matched to *Pseudomonas* (77%).

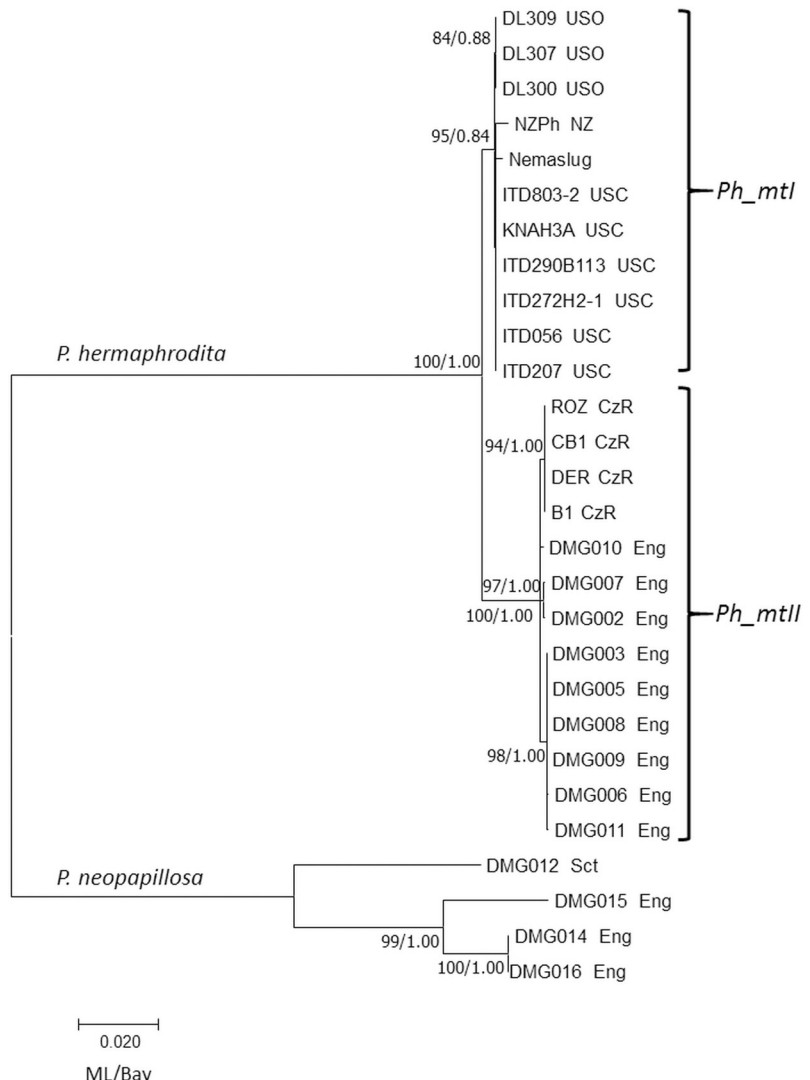

**Fig 2. Maximum likelihood phylogeny of ~2,370 bp of mtDNA from *P. hermaphrodita* and *P. neopapillosa*.** A total of 28 nematode mtDNA sequences, originating from five countries, were analyzed. Country designations are CzR = Czech Republic; Eng = England; NZ = New Zealand; Sct = Scotland; USO & USC = United States of America. *P. hermaphrodita* populations are split into two subclades, Ph_mtI and Ph_mtII. Maximum likelihood and Bayesian phylogenies yielded identical topologies; node support lists maximum likekihood values on left (ML) and Bayesian posterior probabilities are on the right (Bay). The scale bar represents 0.020 substitutions per site.

Five bacterial genera were found associated with the strain DL307, namely *Acinetobacter*, *Brucella*, *Microbacterium*, *Pseudomonas*, and *Ochrobactrum*. For the strain DL309, the three genera *Acinetobacter*, *Pseudomonas*, and *Stenotromophonas* were detected.

*Pseudomonas* bacteria were observed in association with all three strains of nematodes, accounting for 77%, 33%, and 33% of all colonies analyzed in DL300, DL307, and DL309, respectively. Bacteria of the genus *Moraxella* were not found in any of the colonies examined; however, *Acinetobacter*, another genus of the same family as *Moraxella* (i.e. Moraxellaceae) was found associated with both strains DL307 and DL309 (8.3% and 50% of all colonies, respectively). Additional information about the sequence BLASTN matches is provided in S2 Table.

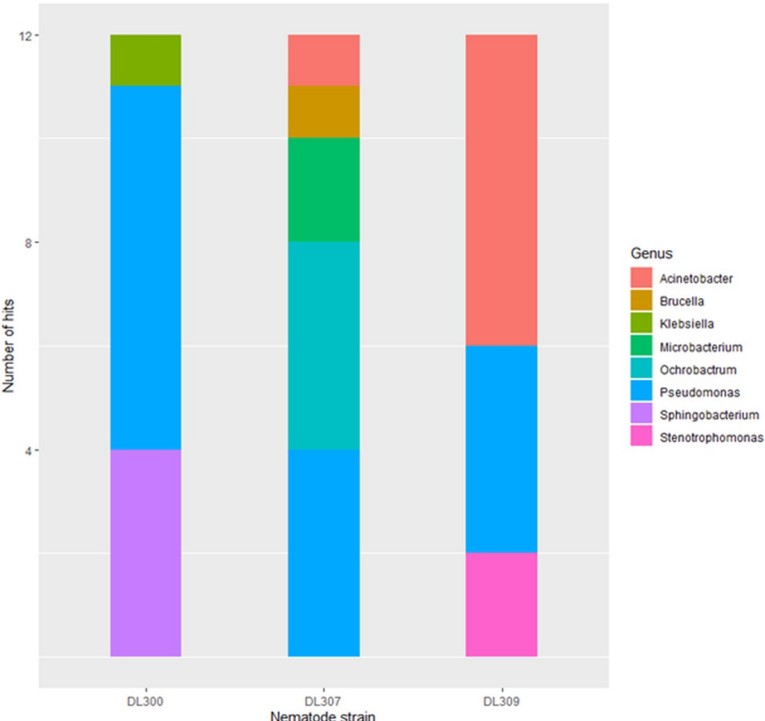

**Fig 3. Bacterial associates of Oregon *P. hermaphrodita*.** The numbers of bacteria identified from different specific genera are indicated by different colors in the bar chart. All three *P. hermaphrodita* strains analyzed were discovered in Oregon (Table 1).

## Discussion

The phylogenetic analyses presented in this study revealed strong support for two major intra-specific mtDNA clades of *P. hermaphrodita* (Figs 1, 2). One clade, Ph_mtI, included all nema-tode strains isolated from slug hosts in North America (California, Oregon) and New Zealand; this clade also included the Nemaslug® strain (derived from a nematode strain isolated in Bristol, England [18]). The other clade, Ph_mtII, included *P. hermaphrodita* strains isolated from slugs in Liverpool, England and Benesov u Prahy and Ceske Budejovice, Czech Republic. These patterns, particularly in Ph_mtI, support the hypothesis that recent discoveries of *P. her-maphrodita* in North America and New Zealand derived from the Nemaslug® strain. Previ-ous sampling efforts (performed in 2007) failed to reveal the presence of *P. hermaphrodita* in California, Oregon and other U.S. states [45], suggesting that the arrival of this biocontrol spe-cies in North America might be very recent [46,47]. But, we cannot rule out an alternative hypothesis that *P. hermaphrodita*, prior to the development of the Nemaslug®, was a globally distributed and panmictic species, and that under sampling in the previous and present studies has given rise to an artifactual and misleading result. Natural populations of the famous model nematode *Caenorhabditis elegans* demonstrate worldwide population-genetic patterns consis-tent with global panmixia, though this pattern is commonly attributed to the effects of human activities [48,49]. The strong and consistent pattern of phylogeographic mtDNA groupings within *P. hermaphrodita*, coupled with very low levels of sequence variation within Ph_mtI (Fig 2), better support the earlier hypothesis that the North American and New Zealand strains originated from the same strain that was developed into Nemaslug®.

Although the three *P. hermaphrodita* strains discovered in Oregon shared a common mtDNA haplotype that was nearly identical to the Nemaslug® haplotype, different genera of

bacteria were found to co-culture with each of these three strains. In our screen, we found only one bacterial genus common to all three nematode samples (*Pseudomonas*), and one other genus that was found in two out of three nematode samples (*Acinetobacter*). For the other six bacterial genera identified, all were unique to a single Oregon nematode strain. Although only limited microbiome inferences are possible from examining small numbers of bacterial colonies able to grow on nutrient agar plates, the patterns suggest that each of the three Oregon *P. hermaphrodita* strains were associated with multiple and distinct bacterial genera at the time of nematode collection. This pattern is consistent with previous research demonstrating that this species associates with diverse bacterial partners [20–22]. These preliminary characterizations of the Oregon nematode microbiomes must be interpreted with caution. An analysis using high-throughput DNA sequencing-based strategies for microbiome characterization will be required to thoroughly understand the nature of microbial communities associated with *P. hermaphrodita*.

It is also noteworthy that we did not identify *Moraxella osloensis*, the bacterial species used in the Nemaslug® commercial formulation. Unlike entomopathogenic nematodes (EPNs) that have a strict symbiotic relationship with bacteria (*Steinernema* sp. with *Xenorhabdus* sp. and *Heterorhabditis* sp. with *Photorhabdus* sp.) that they rely on to kill insects [50], the relationship *P. hermaphrodita* has with bacteria is more complex. Initial studies focused on finding a bacterium that *P. hermaphrodita* could be grown on and produce pathogenic nematodes. Bacteria were isolated from infected slugs and from *P. hermaphrodita* emerging from dead slugs [20,21]. Many different bacterial species were isolated and tested including: *Acinetobacter calcoaceticus*, *Aeromonas hydrophila*, *Aeromonas* sp., *Bacillus cereus*, *Flavobacterium breve*, *FIavobacterium odoratum*, *Moraxella osloensis*, *Providencia rettgeri*, *Pseudomonas fluorescens* (isolate no. 1a), *P. fluorescens* (isolate no. 140), *P. fluorescens* (isolate no. 141), *P. fluorescens* (pSG), *P. paucimobilis*, *Serratia proteamaculans*, *Sphingobacterium spiritocorum* and *Xenorhabdus bovienii*. *Moraxella osloensis* was chosen as it produced consistently high yields of pathogenic nematodes [20,21]. It should be stressed that this bacterium was chosen for commercial production and does not reflect the natural tritrophic interactions that may be occurring between slugs, *P. hermaphrodita* and bacteria in the wild. A study by Rae et al. showed that *P. hermaphrodita*, when grown on rotting slugs or emerging after parasitising slugs (*D. reticulatum*), had no evidence of *M. osloensis* being present [22]. Similarly, Nermut' et al. found that *P. hermaphrodita* strain (DMG0001) lost *M. osloensis* after repeated culturing [51]. Therefore, it is perhaps not too surprising that wild isolated *P. hermaphrodita* from the US do not retain *M. osloensis* as there is little evidence to show this bacterium is vertically transmitted.

Our approach demonstrated the effectiveness of mtDNA in revealing informative phylogenetic patterns within and between species in the genus *Phasmarhabditis*, similar to past successes in other nematode genera [37,52,53]. The mtDNA phylogenetic framework presented in this study for *Phasmarhabditis* spp. (Fig 1) revealed numerous topological consistencies when compared to other recent phylogenetic analyses of this genus that relied on nuclear ribosomal RNA (rRNA) genes [54–60]. For example, our analysis identified *P. huizhouensis* as an early-diverging lineage that is genetically distinct from the other seven species from the genus analyzed here, a pattern consistent with previous analyses of 18S and 28S rRNA loci [57]. Our mtDNA phylogeny revealed an affinity between *P. hermaphrodita* and *P. neopapillosa*, consistent with previous nuclear rRNA studies as well [54]. *P. californica*, a species recently discovered in North America [55], grouped with *Phasmarhabditis bonaquaense*, consistent with previous nuclear rRNA results [54]. *P. papillosa* grouped with *Phasmarhabditis apuliae*, offering a third example of consistency between our mtDNA phylogeny and nuclear rRNA-based analyses [56]. We also note here the discovery of a nematode strain in Oregon (DL308) isolated from *Deroceras reticulatum* that strongly grouped with two other nematode strains (DF5056,

EM434) that were isolated from earthworms [61]. This observation suggests that this unnamed nematode species is capable of associating with different invertebrate hosts.

The Nemaslug® product was originally formulated and commercialized in the United Kingdom, and is available in many European countries. More recently, Dudutech commercialised a similar product under the tradename SlugTech®. However, these biocontrol agents are not commercially available in the United States or New Zealand. Their sale is prohibited by regulatory agencies, primarily due to the unknown impact that *P. hermaphrodita* might have on native gastropods in these countries. Our findings suggest that, despite these regulatory restrictions, nematodes of the same strain as that of the Nemaslug® product have arrived in California, Oregon, and New Zealand. All discoveries of *P. hermaphrodita* in North America and New Zealand, to date, derived from nematodes infecting *Deroceras reticulatum* and other invasive, pest slug species (Table 1). In order to evaluate the potential ecological impacts of these biocontrol-commercialized nematodes, future research is necessary to determine the susceptibilities of New Zealand and North America-native gastropod species to *P. hermaphrodita*. Broader surveys for the nematode, focused on potential native gastropod hosts, are required to assess whether or not *P. hermaphrodita* is currently infecting native gastropod species in North America, New Zealand, and other parts of the world.

## Supporting information

**S1 Fig. Schematic of amplified mitochondrial DNA region for sequence analysis (~5,000 bp).** Linear representation of the fragment of mtDNA amplified using Expand Long Range PCR kit (Roche) with protein-coding genes, ribosomal RNA genes and tRNA genes noted. PCR primers, shown as large arrows, and internal species-specific primers were used for sequencing (see S1 Table). Dashed black lines indicate sequences included in Fig 1 analysis; solid black lines indicate sequences included in Fig 2 analysis.
(TIF)

**S1 Table. PCR and sequencing primers used to target *Phasmarhabditis hermaphrodita* mtDNA.**
(DOCX)

**S2 Table. Bacteria associated with Oregon *P. hermaphrodita*.**
(DOCX)

## Acknowledgments

Thanks to the OSU Center for Genome Research and Biocomputing for DNA sequencing service. Thanks to the following people for assisting with nematode isolation and culturing: Marisa Lutz, Anton Alvarez, Yifei Qiu, Rene Huang, James Cutler, Antoinette Malan, Annika Pieterse and Louwrens Tiedt.

## Author Contributions

**Conceptualization:** Rory J. Mc Donnell, Dee R. Denver.

**Formal analysis:** Dana K. Howe, Anh D. Ha.

**Investigation:** Dana K. Howe, Anh D. Ha, Andrew Colton.

**Methodology:** Dana K. Howe.

**Project administration:** Dana K. Howe.

**Resources:** Dana K. Howe, Andrew Colton, Irma Tandingan De Ley, Robbie G. Rae, Jenna Ross, Michael Wilson, Jiří Nermut, Zhongying Zhao.

**Writing – original draft:** Dana K. Howe, Anh D. Ha.

**Writing – review & editing:** Andrew Colton, Irma Tandingan De Ley, Robbie G. Rae, Jenna Ross, Michael Wilson, Jiří Nermut, Zhongying Zhao, Rory J. Mc Donnell, Dee R. Denver.

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
