## [Decision Letter · Decision Letter 0]

2 Jun 2020

PONE-D-20-06376

Phylogenetic evidence for the invasion of a commercialized European * Phasmarhabditis hermaphrodita * lineage into North America and New Zealand

PLOS ONE

Dear Dr. Howe,

Thank you for submitting your manuscript to PLOS ONE. After careful consideration, we feel that it has merit but does not fully meet PLOS ONE’s publication criteria as it currently stands. Therefore, we invite you to submit a revised version of the manuscript that addresses the points raised during the review process.

We look forward to receiving your revised manuscript.

Kind regards,

Bi-Song Yue, Ph.D

Academic Editor

PLOS ONE

3. In your Methods section, please provide additional details regarding the collected slugs used in your study and ensure you have described the source. For more information regarding PLOS' policy on materials sharing and reporting, see https://journals.plos.org/plosone/s/materials-and-software-sharing#loc-sharing-materials.

4. In your Methods section, please provide additional location information of the collection sites, including geographic coordinates for the data set if available."

5. In your Methods section, please provide additional information regarding the permits you obtained for the work. Please ensure you have included the full name of the authority that approved the collection sites access and, if no permits were required, a brief statement explaining why.

Reviewers' comments:

Reviewer's Responses to Questions

**Comments to the Author**

1. Is the manuscript technically sound, and do the data support the conclusions?

Reviewer #1: Yes

Reviewer #2: Yes

2. Has the statistical analysis been performed appropriately and rigorously? 

Reviewer #1: I Don't Know

Reviewer #2: Yes

3. Have the authors made all data underlying the findings in their manuscript fully available?

Reviewer #1: Yes

Reviewer #2: Yes

4. Is the manuscript presented in an intelligible fashion and written in standard English?

Reviewer #1: Yes

Reviewer #2: Yes

5. Review Comments to the Author

Reviewer #1: The authors have demonstrated similarities between Nemaslug and P. hermaphrodita collected in the USA and NZ. There were differences between the nematodes in this clade and another P. hermaphrodita clade constituted by collections from England and Czech Republic. For these reasons the authors suggest that study has “provided evidence that the nematodes deriving from the Nemaslug biocontrol product have invaded parts of the world where its use is prohibited by regulatory agencies and not commercially available.” The bacterium associated with Nemaslug was not detected in P. hermaphrodita strains collected in the USA.

I tend to agree with the authors’ conclusion. However, I might have made the conclusion a bit more tentative. The sample size of P. hermaphrodita is small, with collections from only four countries. The absence from collections of P. hermaphrodita from the USA and NZ of the bacterium associated with Nemaslug weakens the conclusion. Is there any type of statistical test by which the authors could test their conclusion?

The manuscript is refreshingly clear and therefore easy to read.

Reviewer #2: This is a well written and presented manuscript on an important and critical topic related to the environmental and ecological impacts of deployment of non-native biological control agents of slug pests. The literature is rife with the unexpected and deleterious consequences of deployment of non-native organisms for control of pest species. Here the authors present compelling phylogenetic data to support the idea that Phasmorhabditis hermaphrodita has been spread from its native region (Europe, registered as NemaSlugTM) to New Zealand and North America by deployment of a commercial product unregistered in these areas. This is not only worrisome from a regulatory standpoint, but could have grave ecological impacts on native, non-target gastropods species. The methods employed by the authors are robust and the use of mtDNA as a tool was a good choice for phylogenetic analysis. Particularly compelling was the observation that P. hermaphrodita has only recently appeared in Oregon (since 2007). Although the authors rightly point out that these data might have not detected a panmictic species, the data presented in this manuscript strongly suggest introduction rather than non-detection. Although the phylogenetic data strongly support the introduction hypothesis, the complete absence of the bacterial symbiont Moraxella is perplexing. Because the commercial NemaSlug products contain Moraxella, it is not clear why there is no evidence of this bacterium from any of the populations evaluated. The authors do explain that P. hermaphrodita can associate with multiple bacterial genera in the wild and in the lab, but this does not explain the total absence of Moraxella in the relatively short time period since putative introduction. In my opinion, the manuscript would be stronger with more discussion of this finding. For example, did the bacterial species found in the P. hermaphrodita samples simply outcompete Moraxella, or was Moraxella unsuited to survive the new environment? Having said that, I believe this is an important manuscript and should be published.

6. PLOS authors have the option to publish the peer review history of their article (what does this mean?). If published, this will include your full peer review and any attached files.

Reviewer #1: No

Reviewer #2: No

---

## [Author Response · Author response to Decision Letter 0]

17 Jul 2020

Thank you for the opportunity to revise our manuscript and address these comments. Below I have included the editor’s and reviewers’ comments in black font with my responses in blue font, for clarity.

Editor’s comments:

and

Thank you for the style templates. I corrected the title, indentations, subheading, and adjusted the formatting for Table 1 to match the examples.

We have competed our GenBank submission and completed the sentence with the accession numbers: MT179848 – MT179854 for 18S sequences; MT472183 – MT472270 for mtDNA sequences. These will be “live” with the acceptance of this manuscript.

3. In your Methods section, please provide additional details regarding the collected slugs used in your study and ensure you have described the source. For more information regarding PLOS' policy on materials sharing and reporting, see

https://journals.plos.org/plosone/s/materials-and-software-sharing#loc-sharing-materials.

Thank you for the suggestion about sharing materials. The collected slugs don’t culture well in the lab and are generally not maintained. Most of the nematodes are available from each individual’s labs (identified in Table 1), but there isn’t a central repository for natural nematode collections.

4. In your Methods section, please provide additional location information of the collection sites, including geographic coordinates for the data set if available."

Many of our collection sites were businesses or private farmers. We would prefer not to include the business names or exact addresses/coordinates of these places surveyed because there is a certain amount of trust involved in these entities allowing us to survey, and we don’t want to betray that by “outing” them.

5. In your Methods section, please provide additional information regarding the permits you obtained for the work. Please ensure you have included the full name of the authority that approved the collection sites access and, if no permits were required, a brief statement explaining why.

We appreciate you pointing out this oversight to us. We added a sentence about permits in the methods. Some locations required them while other locations did not. 

Reviewers' comments:

Reviewer's Responses to Questions

COMMENTS TO THE AUTHOR

1. Is the manuscript technically sound, and do the data support the conclusions?

The manuscript must describe a technically sound piece of scientific research with data that supports the conclusions. Experiments must have been conducted rigorously, with appropriate controls, replication, and sample sizes. The conclusions must be drawn appropriately based on the data

presented.

Reviewer #1: Yes

Reviewer #2: Yes

2. Has the statistical analysis been performed appropriately and rigorously?

Reviewer #1: I Don't Know

Reviewer #2: Yes

3. Have the authors made all data underlying the findings in their manuscript fully available?

The PLOS Data policy [1] requires authors to make all data underlying the findings described in their manuscript fully available without restriction, with rare exception (please refer to the Data Availability Statement in the manuscript PDF file). The data should be provided as part of the manuscript or its supporting information, or deposited to a public repository. For example, in addition to summary statistics, the data points behind means, medians and variance measures should be available. If there are restrictions on publicly sharing data—e.g. participant privacy or use of data from a third party—those must be specified.

Reviewer #1: Yes

Reviewer #2: Yes

4. Is the manuscript presented in an intelligible fashion and written in standard English?

Reviewer #1: Yes

Reviewer #2: Yes

5. Review Comments to the Author

Please use the space provided to explain your answers to the questions above. You may also include additional comments for the author, including concerns about dual publication, research ethics, or publication ethics. (Please upload your review as an attachment if it exceeds 20,000

characters)

Reviewer #1: The authors have demonstrated similarities between Nemaslug and P. hermaphrodita collected in the USA and NZ. There were differences between the nematodes in this clade and another P. hermaphrodita clade constituted by collections from England and Czech Republic. For these reasons the authors suggest that study has “provided evidence that the nematodes deriving from the Nemaslug biocontrol product have invaded parts of the world where its use is prohibited by regulatory agencies and not commercially available.” The bacterium associated with Nemaslug was not detected in P. hermaphrodita strains collected in the USA.

I tend to agree with the authors’ conclusion. However, I might have made the conclusion a bit more tentative. The sample size of P. hermaphrodita is small, with collections from only four countries. The absence from collections of P. hermaphrodita from the USA and NZ of the bacterium associated with Nemaslug weakens the conclusion. Is there any type of statistical test by which the authors could test their conclusion?

The manuscript is refreshingly clear and therefore easy to read.

Thank you for your helpful comments. We agree that we have a limited number of sample locations where P. hermaphrodita is found, so we revised our abstract to not extend our findings too broadly. As for the lack of Moraxella osloensis, multiple studies have found that it is quickly lost in culturing. Unlike entomopathogenic nematodes, it is not an obligate symbiosis. We added a paragraph with more explanation in the discussion (see pg 19). 

Reviewer #2: This is a well written and presented manuscript on an important and critical topic related to the environmental and ecological impacts of deployment of non-native biological control agents of slug pests. The literature is rife with the unexpected and deleterious consequences of deployment of non-native organisms for control of pest species. Here the authors present compelling phylogenetic data to support the idea that Phasmorhabditis hermaphrodita has been spread from its native region (Europe, registered as NemaSlugTM) to New Zealand and North America by deployment of a commercial product unregistered in these areas. This is not only worrisome from a regulatory standpoint, but could have grave ecological impacts on native, non-target gastropods species. The methods employed by the authors are robust and the use of mtDNA as a tool was a good choice for phylogenetic analysis. Particularly compelling was the observation that P. hermaphrodita has only recently appeared in Oregon (since 2007). Although the authors rightly point out that these data might have not detected a panmictic species, the data presented in this manuscript strongly suggest introduction rather than non-detection. Although the phylogenetic data strongly support the introduction hypothesis, the complete absence of the bacterial symbiont Moraxella is perplexing. Because the commercial NemaSlug products contain Moraxella, it is not clear why there is no evidence of this bacterium from any of the populations evaluated. The authors do explain that P. hermaphrodita can associate with multiple bacterial genera in the wild and in the lab, but this does not explain the total absence of Moraxella in the relatively short time period since putative introduction. In my opinion, the manuscript would be stronger with more discussion of this finding. For example, did the bacterial species found in the P. hermaphrodita samples simply outcompete Moraxella, or was Moraxella unsuited to survive the new environment? Having said that, I believe this is an important manuscript and should be published.

Thank you for your thoughtful comments. We added a paragraph with more explanation in the discussion (see pg 19) to help clarify the lack of Moraxella osloensis in the newly acquired samples. Multiple previous studies have found that it is quickly lost in culturing because, unlike entomopathogenic nematodes, it is not an obligate symbiosis.

---

## [Editor Report · Decision Letter 1]

23 Jul 2020

Phylogenetic evidence for the invasion of a commercialized European * Phasmarhabditis hermaphrodita * lineage into North America and New Zealand

PONE-D-20-06376R1

Dear Dr. Howe,

We’re pleased to inform you that your manuscript has been judged scientifically suitable for publication and will be formally accepted for publication once it meets all outstanding technical requirements.

Kind regards,

Bi-Song Yue, Ph.D

Academic Editor

PLOS ONE

---

## [Editor Report · Acceptance letter]

4 Aug 2020

PONE-D-20-06376R1 

Phylogenetic evidence for the invasion of a commercialized European * Phasmarhabditis hermaphrodita * lineage into North America and New Zealand 

Dear Dr. Howe:

I'm pleased to inform you that your manuscript has been deemed suitable for publication in PLOS ONE. Congratulations! Your manuscript is now with our production department. 

Kind regards, 

on behalf of

Dr. Bi-Song Yue 

Academic Editor

PLOS ONE